# Myeloid Leukemia of Down Syndrome

**DOI:** 10.3390/cancers15133265

**Published:** 2023-06-21

**Authors:** Aikaterini Kosmidou, Athanasios Tragiannidis, Eleni Gavriilaki

**Affiliations:** 12nd Department of Internal Medicine, General Hospital of Kavala, 65500 Kavala, Greece; 22nd Department of Pediatrics, AHEPA University Hospital, Aristotle University of Thessaloniki, 54636 Thessaloniki, Greece; atragian@auth.gr; 3Hematology Department, G. Papanikolaou Hospital, Aristotle University of Thessaloniki, 57010 Thessaloniki, Greece; elenicelli@yahoo.gr

**Keywords:** Down syndrome, myeloid leukemia, acute megakaryoblastic leukemia, transient abnormal myelopoiesis, *GATA1* gene

## Abstract

**Simple Summary:**

Patients with Down Syndrome have been thoroughly studied over the past 100 years, and many attempts have been made to attain insight into the developmental biology of DS. Given the association of DS with several hematological disorders, it was more than appealing to us to conduct a literature research to identify the rare subtype of acute myeloid leukemias associated with DS -Myeloid Leukemia of Down Syndrome- to investigate its occurrence, clinical presentation, and typical characteristics in terms of blast morphology and immunophenotype, and suggest optimal criteria for early diagnosis and progression monitoring. Among others, the multistep clonal evolution process is being analyzed here, while challenges on treatment of those patients are presented in detail. We suggested that a standardized holistic approach of care for children with Myeloid Leukemia of Down Syndrome should be ensured and applied to provide more enhanced outcomes to those patients.

**Abstract:**

Myeloid leukemia of Down syndrome (ML-DS) is characterized by a distinct natural history and is classified by the World Health Organization (WHO) as an independent entity, occurring with unique clinical and molecular features. The presence of a long preleukemic, myelodysplastic phase, called transient abnormal myelopoiesis (TAM), precedes the initiation of ML-DS and is defined by unusual chromosomal findings. Individuals with constitutional trisomy 21 have a profound dosage imbalance in the hematopoiesis-governing genes located on chromosome 21 and thus are subject to impaired fetal as well as to neonatal erythro-megakaryopoiesis. Almost all neonates with DS develop quantitative and morphological hematological abnormalities, yet still only 5–10% of them present with one of the preleukemic or leukemic conditions of DS. The acquired mutations in the key hematopoietic transcription factor gene *GATA1*, found solely in cells trisomic for chromosome 21, are considered to be the essential step for the selective growth advantage of leukemic cells. While the majority of cases of TAM remain clinically ‘silent’ or undergo spontaneous remission, the remaining 20% to 30% of them progress into ML-DS until the age of 4 years. The hypersensitivity of ML-DS blasts to chemotherapeutic agents, including but not limited to cytarabine, and drugs’ increased infectious and cardiac toxicity have necessitated the development of risk-adapted treatment protocols for children with ML-DS. Recent advances in cytogenetics and specific molecular mechanisms involved in the evolution of TAM and ML-DS are reviewed here, as well as their integration in the improvement of risk stratification and targeted management of ML-DS.

## 1. Introduction

Trisomy 21, the presence of a supernumerary chromosome 21, has been first described as the genetic cause of Down syndrome (DS) in 1959 [1]. Today, it is well established that DS constitutes one of the most frequent and genetically complex chromosomal disorders that are compatible with post-term survival [2]. In the majority of cases, constitutive trisomy 21 results from nondisjunction of Homo sapiens chromosome 21 (HSA21) during meiosis. The presence of all or a portion of a third copy of HSA21 is identified among the most common forms of mental retardation and is thought to be the cause of other medical conditions that affect multiple body systems [2]. The DS phenotype has been estimated to occur in approximately one in every 700 infants in the United States, with an overall prevalence of 10 per 10,000 births in Europe [3]. Due to raised awareness on the maternal-age-related higher risk of developing the syndrome, this prevalence is now decreasing [4]. Although life expectancy of individuals with DS has been extremely improved during recent decades (50–55 years), mortality among them remains approximately 5–10 times higher than mortality in the general population [3]. Patients with constitutional trisomy 21 present with a distinct collection of multiple clinical manifestations, as the extra genetic material results in the phenotypic expression of specific facial characteristics, as well as various malformations, congenital heart defects, immune and endocrine dysfunction, visual and hearing impairment, hematological abnormalities, and Alzheimer’s disease [4].

Almost all neonates with DS develop quantitative and morphological hematological disorders, yet still only 5–10% of them present with one of the preleukemic or leukemic conditions associated with DS [3]. Myeloid leukemia of Down syndrome (ML-DS) is identified as a distinct form of acute myeloid leukemia (AML), with a 46- to 83-fold increased incidence in DS children [5]. It is characterized by a long preleukemic stage, which is called transient abnormal myelopoiesis (TAM). It is marked by the presence of mutations on the *GATA1* gene and is defined by its myeloproliferative nature affecting the megakaryocytic and erythroid lineages [6]. Both TAM and ML-DS present with the clinical and hematologic features of acute megakaryoblastic leukemia (AMKL) and have similar biologic behavior between them that is independent of the blast cell count [5]. AMKL is a rare subtype of AML and is classified as M7, according to the French–American–British (FAB) system. AMKL is defined as a form of leukemia with >20% blasts, of which 50% or more are of the megakaryocyte lineage, and it is associated with extensive myelofibrosis [7]. It is characterized by the presence of megakaryocytic antigens demonstrated by flow cytometry and immunohistochemistry [8]. In terms of blast morphology and immunophenotype, M7 blasts often resemble lymphoblasts, while one or more platelet glycoproteins are expressed on megakaryoblasts, namely CD41 (glycoprotein IIb/IIIa) and/or CD61 (glycoprotein IIIa) [8].

DS children have a 50-fold increased incidence of acute leukemia during the first 5 years of life. The acute leukemias in approximately 60% of affected DS children are myeloid, with at least 50% of these being AMKL [6]. The median age of patients with TAM is 1–1.8 years [4]. DS children with ML–DS have better prognosis compared with non-DS children with a myeloid neoplasm, with 5-year overall survival (OS) of 89–93% [6]. In most cases, TAM spontaneously resolves within 1 to 2 months after birth [5]. However, a percentage of 20–30% of TAM infants will develop ML-DS within the first 4 years of life [6]. It is important to identify and accurately diagnose TAM cases, which has proven challenging due to the absence of clear clinical, hematologic and molecular diagnostic criteria for this condition.

The World Health Organization (WHO) Classification of Tumors of Hematopoietic and Lymphoid Tissues defines TAM as the ‘increased peripheral blood blast cells in a neonate with Down syndrome’, without specifying the percentage blast count considered abnormal [9]. TAM diagnosis requires the presence of *GATA1* mutations together with increased blasts and/or clinical features suggestive of TAM in a neonate with constitutional trisomy 21 [5,6,10]. Regarding TAM clinical presentation, this varies from multiple severe and life-threatening manifestations to asymptomatic disease where TAM may be diagnosed by various nonspecific blood count abnormalities. These include reduced platelet and increased leukocyte counts, as well as peripheral blood blasts [6]. Examination of a peripheral blood film is suggested to be made during the first week of life, as the percentage of circulating blasts often falls rapidly after this period [10]. TAM cells, from their origin in fetal liver, spread throughout the body, infiltrating the liver, pleural and pericardial spaces, skin and, in some cases, bone marrow [10]. This presents as hepatomegaly/hepatopathy (raised transaminases with conjugated hyperbilirubinaemia), as malignant effusions in pericardial and pleural spaces, and as a skin rash due to deposits containing TAM cells in the skin [10]. These and other clinical features—namely splenomegaly found in 30% of cases often because of portal venous obstruction, organomegaly, hepatic fibrosis, hyperleukocytosis, coagulopathy, and multi organ failure—are not entirely specific to TAM, because each of these features may also be found in non-TAM cases [10]. Thus, it is possible that asymptomatic TAM may not be diagnosed in some neonates, while in others, TAM may be overdiagnosed.

To overcome this limitation, many studies have attempted to define clinical and/or hematologic criteria for diagnosis of TAM. Among them, the Oxford Imperial Down Syndrome Cohort Study (OIDSCS) recruited 200 DS neonates to systematically examine their blood findings and suggested that a threshold of >10% peripheral blood blasts in the first week of life, together with clinical features indicative of TAM in a child with DS or mosaic trisomy 21, identifies all neonates with the condition [11]. However, TAM diagnosis also requires the presence of *GATA1* mutations, a genetic change unique to TAM and ML-DS, which is the reason OIDSCS guidelines recommend that in any neonate with a blast percentage > 10% and/or clinical features suggestive of TAM, a peripheral blood sample should be urgently sent for *GATA1* mutation analysis [11].

## 2. Genetic Landscape of Down Syndrome-Related Myeloid Leukemia

TAM is driven by mutations in the hematopoietic transcription factor gene *GATA1*, which result in a truncated isoform (*GATA1s* protein). These mutations are only seen in conjunction with trisomy 21, either constitutional or acquired [11]. This means that even in rare cases of myeloid leukemia in children without DS, the molecular pathogenesis and clinical outcomes are similar to those with DS [6]. These patients acquire trisomy 21, and such cases have already been well documented [12,13]. In addition, GATA1 mutations could be acquired or even germline [6]. All these findings constitute an interesting field of research, as it has been shown that the cooperation between these two lesions—trisomy 21 and *GATA1* mutations—can lead to myeloid proliferation, and the clinical presentation of this condition can be independent of the order of their acquisition [6].

## 3. The Multistep Pathogenesis of ML-DS Development

The pathogenesis of ML-DS is considered to be the result of a multistep clonal evolution process. In the first place, individuals with trisomy 21 have a profound dosage imbalance in the hematopoiesis-governing genes located on chromosome 21 and thus are subject to impaired fetal as well as neonatal erythro-megakaryopoiesis [4]. When these individuals acquire *GATA1* mutations, hematopoietic deregulation is observed, and a selective growth advantage to the leukemic cells is promoted [4]. Further transformation to ML-DS happens when additional mutations affecting chromatin and epigenetic regulators, as well as signaling mediators, contribute to further clonal evolution (Figure 1) [14].

### 3.1. Trisomy 21

It is well established that gains and losses of entire chromosomes or specific genomic regions are hallmarks of cancer. Neonates with gain of chromosome 21 are considered to be more frequently susceptible to hematological malignancies. Constitutive trisomy 21 results in altered hematopoiesis and dysregulated development of the megakaryocytic, erythroid and B-cell lineages [6]. Several studies from genetically engineered models have shown perturbed fetal hematopoiesis in DS individuals and have allowed for the identification of chromosome 21 dosage-sensitive genes [15]. It has been revealed that there is an increased proportion of hematopoietic stem cells (HSCs) and megakaryocyte–erythroid progenitors (MEP) in trisomy 21 fetal livers, which in cooperation with *GATA1s* expression, results in promoting blast and megakaryocyte expansion [15]. Moreover, studies have established fetal hematopoiesis from trisomic DS-derived induced pluripotent stem cells (iPSCs), which lead to significant expansion of myeloid and megakaryocytic progenitors compared with disomic cells [15]. More importantly, *GATA1s* expression correlates with defective embryonic hematopoiesis with a strong bias toward the myelo-megakaryocytic compartment and results in the development of TAM during fetal life [14].

The genes on chromosome 21 have been identified to play a predominant role in myeloid differentiation, and their encoded molecules are the following: transcription factors (ETS-related gene (*ERG*), ETS proto-oncogene 2 (*ETS2*), runt-related transcription factor 1 (*RUNX1*), SON DNA and RNA binding protein (*SON*)), signaling effectors (dual specificity tyrosine phosphorylation regulated kinase 1A (*DYRK1A*), regulator of calcineurin 1 (*RCAN1*)), epigenetic modulators (chromatin assembly factor 1 subunit B (*CHAF1B*), high mobility group nucleosome binding domain 1 (*HMGN1*)), and a subset of miRNAs [14,15,16].

#### 3.1.1. Transcription Factors

*ERG* and *ETS2* are ETS-transcription factors and megakaryocytes oncogenes [6]. The involvement of *ERG* in leukemogenesis is facilitated by its overexpression, which causes megakaryoblastic expansion [6]. On the other hand, ETS2’s role is to promote and regulate megakaryopoiesis. Both *ERG* and *ETS2* strongly cooperate with *GATA1s* protein to drive TAM and/or ML-DS development [14,15]. *RUNX1* transcription factor is involved in the regulation of megakaryocytic differentiation. *RUNX1* accelerates early hematopoiesis in the context of trisomy 21, and in correlation with *ERG*, *ETS2* and *GATA1s* promote leukemogenesis [15,17]. *SON* is a transcription-factor-encoding gene located on HSA21, which plays a crucial role in proper blood cell formation, as it regulates the repression of megakaryocytic differentiation [18].

#### 3.1.2. Signaling Effectors

The signaling molecule *DYRK1A* participates in multiple cellular functions through phosphorylation of the nuclear factor of activated T-cells (NFAT) and other substrates [6]. When the *DYRK1A* gene is overexpressed in murine models, the suppression of NFAT leads to increased megakaryoblastic proliferation [15]. In a corresponding manner, overexpression of *RCAN1*—an endogenous calcineurin inhibitor—represses the calcineurin-NFAT pathway and promotes excessive megakaryopoiesis [14,19].

#### 3.1.3. Epigenetic Modulators

An epigenetic modulator coded by a gene on HSA21 is called *CHAF1B*. This is essential for normal hematopoiesis, and its overexpression impairs myeloid differentiation and promotes myeloid leukemia through binding of chromatin and interference with transcription factors such as *CEBPA* [20]. Another epigenetic modulator is *HMGN1*, which is the chromatin accessibility regulator. When upregulated, *HMGN1* blocks myeloid differentiation and increases clonal progenitor expansion [21].

### 3.2. GATA1 Mutations

The *GATA1* gene is encoded on the X chromosome. It is expressed in erythroid, megakaryocytic, eosinophilic, and mast cells, and it serves as a requirement for the proper growth and maturation of erythroid cells and megakaryocytes [4]. In the case that someone acquires *GATA1* mutations, their megakaryocytes proliferate excessively and do not generate functional platelets [4]. Over one hundred *GATA1* mutations have been reported in individuals with DS, which mainly constitute insertions, deletions, or duplications [15]. In the majority of cases, they occur in exon 2 of the *GATA1* gene and lead to a short isoform of the *GATA1* protein (*GATA1s*), which lacks the amino-terminal activation domain and subsequently creates an early stop codon [6]. Given that the *GATA1* gene plays an essential role in the development of erythroid and megakaryocytic lineages, it is understandable that *GATA1* mutations drive several functional and molecular changes on the regulation of erythro-megakaryocytic progenitors, namely progenitor expansion and disruption of erythroid and megakaryocytic differentiation [6,13]. In the context of trisomy 21, *GATA1* mutations are strengthened and become sufficient to lead to TAM and/or ML-DS, whereas these mutations are not detected when TAM resolves [6]. The unique cooperation between *GATA1* mutations and trisomy 21 in the evolution of TAM has been approved by the fact that *GATA1* mutations have, to date, been discovered in nearly all patients with TAM and ML–DS. The lack of detected *GATA1* mutations in ML-DS individuals is thought to be due to technical and sample limitations [15]. On the other hand, synergy between *GATA1* mutations and subsequent additional chromosomal and genetic alterations is vital for the transformation of TAM to ML-DS, without knowing the exact features that would predominantly predict such transformation [13,14]. Some of the predictors used in clinical practice to decide the possibility of TAM progression to ML-DS include the detection of minimal residual disease (MRD) by flow cytometry (blasts > 0.1%), the persistence of *GATA1* mutations beyond 12 weeks from the initial diagnosis, and the presence of thrombocytopenia (platelet count < 100 × 10^9^/L) [6].

### 3.3. Additional Mutations

Additional somatic mutations are considered to affect genes encoding the cohesin complex, JAK kinases, and epigenetic regulators. Such additional mutations are rarely detected in TAM cases, in which patients carry only *GATA1* mutations. It is suggested that ML-DS progression is mainly caused by activated signaling pathways in cooperation with deregulated epigenetic processes [6].

#### 3.3.1. Cohesin Complex and Associated Components

The cohesin complex is a multi-subunit complex that surrounds chromosomal DNA and regulates its functions, namely sister chromatid cohesion, chromatin remodeling, transcriptional regulation, and DNA damage repair [6]. The mutations affecting the cohesin complex’s core subunits and its modulators are observed with a high occurrence in myeloid malignancies, and they cooperate with *GATA1* mutations and the background of trisomy 21 to drive ML-DS [22]. The cohesin complex’s modulators *STAG2* and *RAD21* have been described to be highly prevalent in ML-DS [4,6]. Studies on DS-derived human iPSCs have shown that the consecutive introduction of *GATA1* and *STAG2* mutations in iPSCs lines result in the disruption of megakaryocyte differentiation and the expansion of the megakaryocytic population [22]. *STAG2* deletion shows as a consequence a decrease in cell growth and proliferation and an increase in cell invasion and metastasis [6,23].

#### 3.3.2. Signaling Pathways

It has been established that the majority of the mutations affecting and activating signaling pathways occur in genes encoding *JAK* family kinases, with the most commonly mutated genes being *JAK2* and *JAK3* [22]. Specifically, the expression of *JAK3* genes is made in myeloid and lymphoid cells, and their mutations present with higher frequency in ML-DS individuals [6]. *JAK* kinases mediate downstream of thrombopoietin (TPO) and granulocyte–macrophage colony-stimulating factor (GM-CSF), and the JAK-STAT signaling pathway controls molecular and cellular processes, such as cell proliferation, differentiation, apoptosis, inflammation, and blood production [22]. In the presence of *JAK3* mutations, normal TPO-mediated *STAT5* activation is inhibited; thus, megakaryopoiesis is dysregulated and repressed [6,22]. The sequence of these events consecutively contributes to leukemogenesis.

Mutations in *RAS* oncogenes (*KRAS*, *NRAS*, *NF1*, *PTPN11*) have been associated with uncontrolled cell growth and colony formation, without sufficient clarity to how the mutated genes mediate leukemogenesis and cooperate with trisomy 21, *GATA1* mutations and other additional mutations in the cohesin complex or the epigenetic regulators to drive ML-DS [4].

#### 3.3.3. Epigenetic Regulators

Multiple epigenetic regulators have been described to contribute to leukemogenesis [6]. Among them, the most commonly presented are: additional sex combs-like 1 (*ASXL1*), BCL6 corepressor (*BCOR*), *DNMT1*, *DNMT3A*, embryonic ectoderm development (*EED*), E1A binding protein P300 (*EP300*), *EZH2*, KAT8 regulatory NSL complex subunit 1 (*KANSL1*), lysine demethylase 6A (*KDM6A*), lysine methyltransferase 2C (*KMT2C*), N-acetyltransferase 6 (*NAT6*), SUZ12, and tet methylcytosine dioxygenase 2 (*TET2*) [22,23]. *EZH2* acts together with *SUZ12*, and they form the polycomb repressive complex 2 (*PRC2*). Both *EZH2* and *SUZ12* work as tumor suppressors and chromatin modifiers; hence, the mutated loss of their function and subsequent lack of PRC2 subunits’ function lead to blockage of megakaryocytic differentiation and acceleration of megakaryocytic proliferation [22,23].

Despite all these advances in understanding the underlying mechanisms associated with ML-DS presentation, there is still debate on the exact treatment plan for these patients, while little is yet known on the relapse of the disease, and the management of relapsed patients remains challenging [14].

## 4. Risk Factors for Early Death

ML-DS death is predominantly associated with progressive hepatopathy with cholestasis, leading to hepatic fibrosis, subsequent disseminated intravascular coagulation (DIC) and multiorgan failure [10]. If this is not the case, non-hepatic deaths mainly constitute the result of cardiorespiratory failure associated with malignant pericardial and pleural effusions, hydrops fetalis, renal failure and severe liver disease [10]. Given that DS neonates often present with congenital heart disease (CHD), early death can also occur secondary to heart complications and is not directly attributable to ML-DS [10]. Several clinical studies have examined DS children suffering from ML-DS and have tried to decide clinical and/or hematologic factors predictive for early death. The factor found to be most consistently associated with early death among three large studies was hyperleukocytosis (white blood cell (WBC) count > 100 × 10^9^/L) [24,25,26].

The Children’s Oncology Group (COG) enrolled patients with TAM and studied them until resolution of the disease or until the time of development of ML-DS [26]. During that period, patients were examined for the presence of life-threatening symptoms (LTS) and were enrolled in either the observation or intervention arms of the study according to the severity of presenting signs and symptoms, with the aim of determining patients with TAM eligible for intervention [26]. LTS, the sole criteria for intervention in TAM patients, was defined as one or more of the following: signs of extreme leukocytosis, hyperviscosity, blast count > 100,000/μL, hepatopathy, ascites or massive hepatomegaly, hepatosplenomegaly causing respiratory or feeding compromise, heart failure that is not directly the result of a congenital heart defect, hydrops fetalis, renal failure, or DIC with bleeding [26].

## 5. Clinical Outcomes and Treatment Plan of ML–DS

There are several clinical studies published that have prospectively collected data on TAM and myeloid leukemia in neonates with DS. Such studies have demonstrated that the majority of TAM patients undergo spontaneous remission, while their findings have indicated that the persistence of MRD can be used in clinical practice to predict risk for developing ML–DS [27,28]. OIDSCS guidelines suggest that there is no evidence for the routine use of therapeutic drugs in neonates solely to prevent later development of ML-DS [10]. Although the early use of chemotherapy in TAM patients has not yet proven its beneficial role in preventing the progression to ML-DS, TAM patients who suffer from LTS are becoming involved in chemotherapy protocols in an effort to reduce the severity of their symptoms rather than eradicate the clone [4,26].

### 5.1. Enhanced Cytarabine Sensitivity in Patients with ML-DS

With regard to the ML-DS treatment plan, it has been established that children with ML-DS have generally better clinical outcomes than non-DS children with myeloid leukemia [6]. This fact could possibly be explained by the high sensitivity of ML-DS blasts to chemotherapy; thus, patients present improved outcomes [14]. Several clinical studies have been conducted in the past to compare DS and non-DS patients with myeloid leukemia, their blasts’ metabolism, and their response to therapy, and based on their findings, the observed sensitivity of DS myeloblasts was approximately 10-fold more enhanced to cytarabine compared with non-DS myeloblasts [28,29]. This higher sensitivity is thought to be secondary to the increased expression of the chromosome 21-localized gene CBS and the potential mechanisms that enhance the susceptibility of cells to undergo apoptosis [4].

### 5.2. Challenges on Treatment of ML-DS Patients

Over the past 30 years, ML-DS children have been either undertreated or registered on treatment protocols for non-DS myeloid neoplasms. The results were either high rates of treatment failure or—in cases of successful outcomes in therapy—a high frequency of treatment-related mortality and/or infectious morbidity [15]. The Acute Myeloid Leukemia–Berlin Frankfurt Münster (AML-BFM) study group reported microbiologically documented infections in 30% of patients, the majority of which were Gram-positive bacteria, suggesting that optimizing drug dosing may improve outcomes while reducing toxicity [30]. Unfortunately, the optimal balance between dose intensity and the risk of treatment toxicity has not been determined. More than 40% of the infectious complications occur in induction therapy, which is known to be a vulnerable phase for patients. Treatment protocols for children with ML-DS that include reduced dose intensity therapy have resulted in a significant decrease in microbiologically documented infections [30]. In search of the main causes for the increased susceptibility of DS children in infections, it is assumed that some non-immunologic factors associated with dysmorphic features and anatomical abnormalities (anatomical abnormalities of the airways, congenital anomalies of the lower respiratory tract, obstructive sleep apnea, gastroesophageal reflux disorders) in affected patients could possibly contribute to the increased frequency of infections in patients with DS. CHD is one of the main problems leading to high morbidity in approximately 40% of patients. This fact could also suggest the higher risk of serious infections such as pneumonia and sepsis in patients with DS with CHD [30]. Moreover, major immune disorders in DS children have been observed (decreased neutrophil chemotaxis, decreased number of NK cells, decreased absolute number of monocytes, decreased naive T cells, lack of memory cell formation, smaller size of thymus, inadequate body response to vaccination) and are thought to play a crucial role in infection susceptibility in that population [30].

### 5.3. Clinical Trials

Children with ML-DS have been enrolled in uniform ML-DS specific protocols with outcomes significantly better using reduced-intensity ML–DS therapeutic agents, including daunorubicin, etoposide, and intrathecal cytarabine [15,31,32]. However, high-dose cytarabine was established as an important component of therapy, with early administration leading to improved outcomes. Several attempts of dosage reduction in standard-risk patients have resulted in significantly lower event-free survival (EFS) [32,33]. According to clinical trials conducted in children with DS and myeloid leukemia, there has been significant progress in recent years in the treatment plan of these children. The outcomes of the most important trials in ML-DS are summarized in Table 1.

The COG AAML1531 study was designed to introduce risk stratification of treatment intensity for ML-DS, based on the previously identified prognostic factor, MRD, at the end of the first induction course [33]. Patients were stratified as standard-risk (SR) patients, identified by negative MRD using flow cytometry (<0.05%) and did not receive the historically administered high-dose cytarabine course (HD-AraC), while high-risk (HR) patients were directed to more intense consolidation. Specifically, all patients received the same first course induction I (thioguanine 50 mg/m^2^/dose twice daily, days 1–4; cytarabine 200 mg/m^2^ per 24 h continuous infusion, days 1–4; daunorubicin 20 mg/m^2^ on days 1–4 over 1–15 min) and a single dose of age-based intrathecal cytarabine. Standard-risk therapy consisted of two more courses of thioguanine, cytarabine, and daunorubicin followed by two identical courses of intensification therapy (intensification I and II: cytarabine 100 mg/m^2^ per 24 h continuous infusion, days 1–7; etoposide 125 mg/m^2^ per day, days 1–3). Intensification course I for the high-risk arm included cytarabine at 33 mg/kg per day, days 1–5 (ten doses) and etoposide at 5 mg/kg per day, days 1–5 (five doses), whereas intensification course II included cytarabine at 33 mg/kg per day, days 1, 2, 8, 9 (eight doses) and L-asparaginase at 200 U/kg per day, days 2, 9 (two doses). The results have shown that omission of HD-AraC led to a statistically and clinically significant decrease in 2-year EFS from 93.5% to 85.6% (95% confidence interval (CI), 75.7–95.5). OS at 2 years was 91.0% (95% CI, 83.8–95.0). More importantly, negative MRD in the COG AAML1531 study did not identify a favorable risk group for whom HD-AraC was dispensable.

The Nordic Society for Pediatric Hematology and Oncology (NOPHO), Dutch Childhood Oncology Group (DCOG), and AML-BFM study groups analyzed the outcomes of 170 pediatric patients with ML-DS and compared them with the historical control arm, in which reduced-intensity protocol was used for ML-DS patients [32]. The treatment consisted of four cycles of polychemotherapy: first cycle (cytarabine 100 mg/m^2^ per day, days 1–2 and 100 mg/m^2^ per 12 h, days 3–8, idarubicin 8 mg/m^2^ per day, days 3, 5, and 7, and etoposide 150 mg/m^2^ per day, days 6–8); second cycle (cytarabine 500 mg/m^2^ per day, days 1–4 and idarubicin 5 mg/m^2^ per day, days 3 and 5); third cycle (cytarabine 1 g/m^2^ per 12 h, days 1–3 and mitoxantrone 7 mg/m^2^ per day, days 3–4); and fourth cycle (high-dose cytarabine 3 g/m^2^ per 12 h, days 1–3). It was found that 5-year OS was 89% ± 3% vs. 90% ± 4%, whereas 5-year EFS was 87% ± 3% vs. 89% ± 4%. In other words, outcomes between the two groups have been proven comparable, and the study group concluded that reducing intensity of therapy did not impair prognosis in ML-DS compared with the historical control.

The COG AAML0431 trial enrolled 204 eligible patients and consisted of four cycles of induction and two cycles of intensification therapy based on the treatment plan of previous trials [31]. HD-araC was used in the second induction cycle instead of the intensification cycle, and one of four daunorubicin-containing induction cycles were eliminated. The treatment consisted of four cycles of induction therapy and two cycles of intensification therapy. Induction cycles I, III, and IV consisted of continuous-infusion araC 6.7 mg/kg per day for 4 days (96 h), continuous-infusion daunorubicin 0.67 mg/kg per 24 h for 4 days (96 h), and oral 6-thioguanine 1.65 mg/kg twice daily for 4 days. Induction cycle II consisted of araC 100 mg/kg administered as a 3 h infusion every 12 h for four doses on days 1 and 2 and repeated on days 8 and 9 (total eight doses), with Escherichia coli asparaginase (200 U/kg) being administered intramuscularly 3 h following the last dose of araC on days 2 and 9. Intensification cycles I and II consisted of continuous-infusion araC 3.3 mg/kg per 24 h for 7 days (168 h) and etoposide 4.2 mg/kg per dose administered as a 1 h infusion for 3 days. The trial’s findings indicated that 5-year OS was 93.0% and EFS was 89.9%, suggesting that earlier use of HD-araC led to better EFS and OS.

## 6. Conclusions and Future Directions

Bearing in mind the absence of uniformly defined high-risk criteria for therapy of individuals with ML-DS, the goal of the next-generation trials would be to identify the optimal dose and schedule for cytarabine treatment and to provide therapy risk stratification. Apart from this, new molecular targets for prevention and treatment are now starting to be unraveled, regarding the underlying mutations in signaling pathways and epigenetic processes. Given that the overactivation of pathways, such as the JAK-STAT pathway, is predominantly involved in ML-DS pathogenesis, inhibition of these pathways, e.g., using *JAK1/2* inhibitors, may play a crucial role in the therapeutic approach of ML-DS patients with activating JAK-STAT mutations [22]. For JAK/STAT mutations, the *JAK1/2* inhibitors, ruxolitinib and momelotinib, have been investigated and are now FDA-approved for myelofibrosis [38], and researchers remain optimistic that their use in myelofibrosis can pave the way for ML-DS treatment as well. Additionally, drugs targeting the *RAS* oncogenes and PI3K/PKB signaling pathway could be used as potential therapeutic targets for patients with such mutations.

Furthermore, targeting cohesin-complex-mutated cells could serve as a treatment strategy and could lead to optimal treating outcomes [6], either by directly modulating the cohesin complex’s subunits and its regulators or by targeting altered DNA damage repair mechanisms [39].

Finally, three chromosome 21 miRNAs (miR-99a, miR-125b and miR-155) were found to be overexpressed in blast cells from ML-DS, and their blockage could inhibit *GATA1s*-induced leukemia development [16]. Hence, inhibitors of these overexpressed pathways could be trialed so that miRNAs become potential therapeutic targets in the future.

In any of these cases, consideration should be given to a unified international protocol, which will provide more enhanced outcomes for patients with ML-DS and permit a greater number of questions to be answered.

## 7. Summary

Three main genetic steps constitute the sequence of pathogenetic events leading to ML-DS: trisomy 21, mutations in *GATA1* gene, and secondary mutations in transcriptional regulators and signaling pathways. While it is clear that the impaired hematopoietic differentiation of trisomy 21 serves as the basis for the selective advantage of clones with *GATA1* mutations, it is unknown under which mechanisms the additional mutations promote the development of ML-DS. Additionally, it has yet to be answered whether the specific combination and order in which mutations are acquired is critical and decisive for the progression of ML-DS in children and their response to treatment. Recent application of CRISPR/Cas9 technology in iPSC-based models of ML-DS has started to provide some essential answers to these questions.

## Figures and Tables

**Figure 1 cancers-15-03265-f001:**
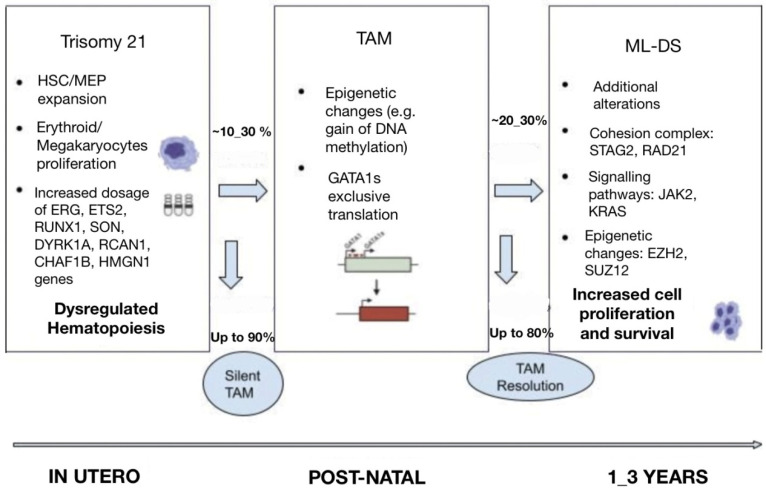
Multi-step pathogenesis of ML-DS.

**Table 1 cancers-15-03265-t001:** Summary of clinical trials for ML-DS children.

Group	Trial	Year	Number of Patients	CR ^1^	EFS ^2^ (%)	OS ^3^ (%)	Relapses (%)	Treatment-Related Mortality (%)
COG ^4^	AAML1531 [33]	2015–2023	256	NA	85.6 (2-year)	91.0 (2-year)	10.8	NA
BFM ^6^/DCOG ^7^/NOPHO ^8^	ML–DS ^5^ 2006 [32]	2006–2015	170	NA ^9^	87.0 (5-year)	89.0 (5-year)	5.3	2.9
COG ^4^	AAML0431 [31]	2007–2011	204	177/202	89.9 (5-year)	93.0 (5-year)	6.9	1
JPLSG ^10^	AML-D05 [34]	2008–2010	72	69/72	83.3 (3-year)	87.5 (3-year)	13.9	1.4
JCCLSG ^11^	AML 9805 [35]	1998–2006	24	21/24	82.6 (5-year)	87.5 (5-year)	4.2	12.5
Japanese Childhood AML	AML99 [36]	2000–2004	72	70/72	83.3 (4-year)	83.7 (4-year)	1.4	12.5
COG ^4^	2971 [37]	1999–2003	132	91/108	79.0 (5-year)	84.0 (5-year)	NA	2.3

^1^ CR, complete remission; ^2^ EFS event-free survival; ^3^ OS, overall survival; ^4^ COG, Children’s Oncology Group; ^5^ ML-DS, myeloid leukemia associated with Down syndrome; ^6^ BFM, Berlin–Frankfurt–Münster study group; ^7^ DCOG, Dutch Childhood Oncology Group; ^8^ NOPHO, Nordic Society of Pediatric Hematology and Oncology; ^9^ NA, not available; ^10^ JPLSG, Japanese Pediatric Leukemia/Lymphoma Study Group; ^11^ JCCLSG, Japanese Children’s Cancer and Leukemia Study Group.

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
