# Peer review of "Myeloid Leukemia of Down Syndrome"

_cancers, 2023, doi:10.3390/cancers15133265_

Round 1

Reviewer 1 Report

In this review, the Recent advances in cytogenetics and specific molecular mechanisms of Myeloid leukemia of Down syndrome (ML-DS) were comprehensively summarized. Moreover, the risk stratification and targeted management of ML-DS were depicted. However, some details are still needed to be revised.

1.     The role of GATA1 mutations in promoting the development of ML-DS was presented in two different sections of the manuscript, and the illustrations on GATA1 mutations are recommended to be further revised.

2.     The authors are suggested to add subheadings to the section which introduced the clinical outcomes and treatment plan of ML–DS.

3.     Further research and clinical trials on novel agents could be added to the manuscript to elaborate the treatment plan of ML–DS.

4.     The authors should supplement detailed treatment strategies which were implemented in clinical trials for ML-DS children.

5.     The formats of references were not consistent.

There are some grammatical mistakes in the article

Author Response

Point 1: The presentation of the role of GATA1 mutations in the development of ML-DS has been revised. The formation of GATA1 mutations is now illustrated only in the 3.2 “GATA1 Mutations” section of the manuscript. Section 2. “Genetic Landscape of Down Syndrome-Related Myeloid Leukemia” is considered as an introductory presentation of the molecular mechanisms of ML-DS, in which the significance of the two major steps in the pathogenesis of ML-DS (trisomy 21 and GATA1 mutations) is emphasized. 

Point 2: Subheadings were added to section 5. “Clinical Outcomes and Treatment Plan of ML-DS”. 

Point 3: Section 6. “Conclusions and Future Directions” was supplemented with more clinical trials and research, regarding some of the therapeutic targets and novel agents. 

Point 4: Details on treatment strategies were added to the section of clinical trials, and the dosages of drugs used were illustrated. 

Point 5: References were further revised.

Reviewer 2 Report

Authors provide a detailed summary of multiple studies linked to role of trisomy 21 in the development of Myeloid leukemia of Down syndrome (ML-DS). Authors have very neatly discussed the distinct characteristics of acute myeloid leukemia from ML-DS. Authors have also thoroughly discussed the role of GATA1 mutation in the multistep pathogenesis of ML-DS development. The review also provides an insight into the role played by other factors including epigenetic and signaling mediators. The review includes latest findings from the field and provides a comprehensive view of molecular pathogenesis of Myeloid leukemia of Down syndrome (ML-DS).

Comments regarding the submission.

1.       The studies cited in this review have been systematically put forward and the outcome of all the studies have been well summarized.

2.       Scientifically correct conclusions have been drawn when combining various independent studies to make a collective scientific statement.

3.       The table included in the review provides an opportunity to a have quick and concise look at the different clinical studies on ML-DS and there outcomes. Such tables should also be included for other sections of the review.

4.       The review article has been neatly written with acronyms and abbreviations properly explained for greater accessibility.

5.       The review is a novel summary of role of genetic, epigenetic and signaling cascades in the pathogenesis of ML-DS.

6.       Overall the review is a good read which provides a critical summary of the topic discussed.

Author Response

Thank you very much for your comments and positive feedback on our review. 

Reviewer 3 Report

This review aims to present current knowledge regarding molecular and clinical advances for myeloid proliferations associated with Down syndrome. I find that the strongest aspect of this review is the last section focusing on the review of the clinical outcomes and clinical trials for myeloid leukemia associated of Down syndrome (ML-DS). However, there are areas that can be improved mainly including the discussion of current knowledge about the pathogenesis and genomic landscape of this entity.  

Main recommendations include:

1. Discuss and/or propose diagnostic criteria for TAM and myeloid leukemia of Down syndrome; discuss about % of blasts and the requirement for the presence of GATA1 mutation 

2. Can the authors clarify the following statement: "Both TAM and ML-DS present with the clinical and hematologic features of acute megakaryoblastic leukemia (AMKL), and have similar biologic behavior and prognosis between them -independent of the blast cell count(5)" . It is not clear what they are comparing/referring to.

3. Is this statement correct? " It is characterized by a long preleukemic myelodysplastic syndrome (MDS)-like stage, which is called transient abnormal myelopoiesis (TAM)" . It can be confusing to use the MDS term for TAM since MDS in pediatric population indicated bone marrow transplantation, while TAM can resolve without treatment. 

4. "Thus, it is possible that asymptomatic TAM may not be diagnosed in some neonates, while in others TAM may be overdiagnosed ". This statement can benefit for providing some opinions/recommendations by the authors. Will they propose GATA1 mutations and in what scenarios? are there studies that they can reference too? 

5. Can the authors clarify this sentence? "That means that even in rare cases of TAM in children without DS, the molecular pathogenesis and clinical outcomes are similar to those with DS(6)." What clinical scenarios are they referring too? I would avoid to use the term TAM outside of the clinical context of Down syndrome. 

The manuscript can be benefit by editing of English language. Just to point out one example, a  sentence in section 3.3.2 that will be benefit from editing "In specific, the expression of JAK3 genes is made in myeloid and lymphoid cells,". 

The use of "GATA1s mutations" is confusing, since the mutations results in the  shorter GATA1s isoform expression. 

Author Response

Point 1: On the fourth paragraph of section 1. "Introduction", the phrase "TAM diagnosis requires the presence of GATA1 mutations together with increased blasts and/or clinical features suggestive of TAM in a neonate with constitutional trisomy 21" was added to emphasize the importance of the two main pathogenetic events (trisomy 21 and GATA1 mutations) and the requirement of their presence in developement and promotion of TAM. On the following sections, the crucial cooperation of these two lesions is analyzed in detail. 

Given that there are no official clinical, hematologic and molecular diagnostic criteria for TAM, we chose to present a detailed description of the multiple clinical manifestations of this condition -which can vary from asymptomatic to life-threatening disease- and give an insight into various nonspecific blood count abnormalities, which can be used in clinical practice to efficiently diagnose TAM cases.

Regarding the percentage of blasts that characterize the disease, the WHO classification of Tumors of Haematopoietic and Lymphoid Tissues defines TAM as the ‘increased peripheral blood blast cells in a neonate with Down syndrome’, without specifying the percentage blast count considered abnormal. Some authors suggest that overt TAM with a % of blasts >10% occurs in approximately 10–15% of DS neonates, but this statement does not constitute a safe criterion for diagnosing TAM.

Point 2: According to the "2016 revision to the WHO classification of myeloid neoplasms and acute leukemias", both TAM and ML-DS are megakaryoblastic proliferations and seem to present with similar behaviour which is independent of the blast cell count, and this is the reason why they are not subclassified as MDS or AML. Instead, they both belong to the uniform category of myeloid proliferations of Down syndrome. Their difference lie on the fact that TAM occurs at birth or within days after birth and in most cases resolves spontaneously, while ML-DS occurs in the first three years of life and persists if remains untreated. 

Point 3: The phrase "myelodysplastic syndrome (MDS)-like stage" was eliminated, as MDS cannot simulate TAM. According to the "2016 revision to the WHO classification of myeloid neoplasms and acute leukemias", ML-DS is categorized as a separate entity without being associated with acute leukemias with myelodysplasia-related changes. In addition, MDS in pediatric population and myeloid proliferations in DS patients have different indications regarding their treatment plan. 

Point 4: The statement "Thus, it is possible that asymptomatic TAM may not be diagnosed in some neonates, while in others TAM may be overdiagnosed " can be explained by the following points:

  • TAM may be asymptomatic or may present with a very broad spectrum of clinical features. No single clinical feature is entirely specific to TAM because each of these features may also occur in the absence of TAM. This is why diagnosis of TAM based only on its clinical manifestations is considered unsafe. Presence of one or more of these features in the absence of a clear alternative explanation should lead to the early consideration of a diagnosis of TAM. 
  • As it has been already mentioned, there is no clear threashold for the % of blasts in diagnosis of TAM. Although a threshold of >10% peripheral blood blasts in the first week of life seems to identify all neonates with clinical features of TAM, some neonates with DS with blasts >10% do not have a GATA1 mutation even when very sensitive (NGS)-based methods are used. Instead, the majority of neonates with blasts >20% are considered to have a GATA1 mutation, hence a higher threshold is likely to be more specific for TAM.  There is a hypothesis that setting a blast threshold of >10% will identify more cases of TAM and that GATA1 mutation analysis is particularly important in neonates with blasts of 10–20% to prevent overdiagnosis of TAM.
  • One or more technical factors could lead to failure to demonstrate GATA1 mutations in clinically suspected TAM cases.
  •  All neonates with high suspicion of TAM should have a full blood count and blood film requested in the first 3 days of life and a formal assessment of the peripheral blood blast cell percentage, as the count of circulating blasts often fall rapidly after this period of time, and TAM cases could be missed this way. 

Point 5: In the manuscript, two cases of patients with acquired (not constitutional) trisomy 21 have been referenced, in whom the myeloproliferative clone was identified. The use of TAM to describe these patients' condition was eliminated and instead the term "myeloid leukemia" was used. This statement has been made to emphasize that the cooperation between these two lesions -trisomy 21 and GATA1 mutations- can lead to myeloid leukemia independently of the order of their acquisition. 

The term "GATA1s mutations" was eliminated. Instead, "GATA1 mutations" has been used.  

Round 2

Reviewer 3 Report

The current version is significantly improved appropriately addressing all reviewers' comments. There are a few minor recommendations suggesting for the revised manuscript: 

1.     Can the authors clarify what they mean by the term similar biological behavior when comparing TAM and ML-DS? Do they mean that the % blast and the presence of GATA1 mutation do not distinguish between the two entities except that TAM is usually not the case in infants older than 1 year of age?

“Both TAM and ML-DS present with the clinical and hematologic features of acute megakaryoblastic leukemia (AMKL), and have similar biologic behavior between them -independent of the blast cell count(5).” 

2.     The FAB classification and M0-M7 terminology are rarely used in the literature. I would propose that the authors re-write the following paragraph summarizing the WHO classification/terminology for AMKL including the main categories for non-Down syndrome AMKL and DS-AMKL. The former includes AML with t (1;22) or other genetic alterations including GLIS2 fusions (see review article from Gruber TA, BLOOD 2015).

“AMKL is a rare subtype of AML and is classified as M7, according to the French-American-British (FAB) system. AMKL is defined as the form of leukemia with >20% blasts, of which 50% or more are of the megakaryocyte lineage, and is associated with extensive myelofibrosis(7). It is characterized by the presence of megakaryocytic antigens demonstrated by flow cytometry and immunohistochemistry(8). In terms of blasts’ morphology and immunophenotype, M7 blasts often resemble lymphoblasts, while one or more platelet glycoproteins are expressed on megakaryoblasts, namely CD41 (glycoprotein IIb/IIIa) and/or CD61 (glycoprotein IIIa)(8). “

3.     Recommend to re-write the following sentence and specifically clarify that >50% of acute leukemias (not myeloid leukemias) in children with DS are AMKL: “DS children have a 50-fold increased incidence of acute leukemia during the first 5 years of life. The acute leukemias in approximately 60% of affected DS children are myeloid, with at least 50% of these being AMKL(6). “

4.     I will recommend changing ”trying to decide” to “investigating” in the following sentence: Several clinical studies have examined DS children suffering from ML-DS, and have tried to decide clinical and/or hematologic factors predictive for early death. 

Only minor editing is recommended as indicated in the comments/recommendations.